Conserved molecular signatures in the spike protein provide evidence indicating the origin of SARS-CoV-2 and a Pangolin-CoV (MP789) by recombination(s) between specific lineages of Sarbecoviruses

Khadka Bijendra 1 2
Gupta Radhey S. gupta@mcmaster.ca 1
1 Department of Biochemistry and Biomedical Sciences, McMaster University , Hamilton , Ontario , Canada
2 Department of Pharmacology and Toxicology, University of Toronto , Toronto , Canada
Bishop-Lilly Kimberly
Electronic publication date: 2021 Nov 12
Publication date: 2021
Volume: 9
Electronic Location ID: e12434
Received 2021 Jun 8; Accepted 2021 Oct 14
Copyright: ©2021 Khadka and Gupta
Copyright year: 2021
Copyright holder: Khadka and Gupta
License: This is an open access article distributed under the terms of the Creative Commons Attribution License, which permits unrestricted use, distribution, reproduction and adaptation in any medium and for any purpose provided that it is properly attributed. For attribution, the original author(s), title, publication source (PeerJ) and either DOI or URL of the article must be cited.
License URL: https://creativecommons.org/licenses/by/4.0/

Keywords: Conserved signature indels (CSIs); Coronaviruses; Molecular signatures; Sarbecoviruses; SARS and SARS-CoV-2-related viruses; Spike protein; Bat CoVZXC21, CoVZC45, PrC31; Pangolin CoV_MP789; Genetic recombination; Origin of SARS-CoV-2 virus

Funding: The Natural Science and Engineering Research Council of Canada x RGPIN-2019-06397 This work was supported by the research grant No. RGPIN-2019-06397 from the Natural Science and Engineering Research Council of Canada awarded to Radhey S. Gupta. The funders had no role in study design, data collection and analysis, decision to publish, or preparation of the manuscript.

==============================
Both SARS-CoV-2 and SARS coronaviruses (CoVs) are members of the subgenus Sarbecovirus. To understand the origin of SARS-CoV-2, sequences for the spike and nucleocapsid proteins from sarbecoviruses were analyzed to identify molecular markers consisting of conserved inserts or deletions (termed CSIs) that are specific for either a particular clade of Sarbecovirus or are commonly shared by two or more clades of these viruses. Three novel CSIs in the N-terminal domain (NTD) of the spike protein S1-subunit (S1-NTD) are uniquely shared by SARS-CoV-2, Bat-CoV-RaTG13 and most pangolin CoVs (SARS-CoV-2r clade). Three other sarbecoviruses viz. bat-CoVZXC21, -CoVZC45 and -PrC31 (forming CoVZC/PrC31 clade), and a pangolin-CoV_MP789 also contain related CSIs in the same positions. In contrast to the S1-NTD, both SARS and SARS-CoV-2r viruses contain two large CSIs in the S1-C-terminal domain (S1-CTD) that are absent in the CoVZC/PrC31 clade. One of these CSIs, consisting of a 12 aa insert, is also present in the RShSTT clade (Cambodia-CoV strains). Sequence similarity studies show that the S1-NTD of SARS-CoV-2r viruses is most similar to the CoVZC/PrC31 clade, whereas their S1-CTD exhibits highest similarity to the RShSTT- (and the SARS-related) CoVs. Results from the shared presence of CSIs and sequence similarity studies on different CoV lineages support the inference that the SARS-CoV-2r cluster of viruses has originated by a genetic recombination between the S1-NTD of the CoVZC/PrC31 clade of CoVs and the S1-CTD of RShSTT/SARS viruses, respectively. We also present compelling evidence, based on the shared presence of CSIs and sequence similarity studies, that the pangolin-CoV_MP789, whose receptor-binding domain is most similar to the SARS-CoV-2 virus, has resulted from another independent recombination event involving the S1-NTD of the CoVZC/PrC31 CoVs and the S1-CTD of an unidentified SARS-CoV-2r related virus. The SARS-CoV-2 virus involved in this latter recombination event is postulated to be most similar to the SARS-CoV-2. Several other CSIs reported here are specific for other clusters of sarbecoviruses including a clade consisting of bat-SARS-CoVs (BM48-31/BGR/2008 and SARS_BtKY72). Structural mapping studies show that the identified CSIs form distinct loops/patches on the surface of the spike protein. It is hypothesized that these novel loops/patches on the spike protein, through their interactions with other host components, should play important roles in the biology/pathology of SARS-CoV-2 virus. Lastly, the CSIs specific for different clades of sarbecoviruses including SARS-CoV-2r clade provide novel means for the identification of these viruses and other potential applications.

Introduction

The current worldwide pandemic is caused by a novel coronavirus (CoV) designated as SARS-CoV-2 (Coronaviridae Study Group of the International Committee on Taxonomy of V, 2020; Zhou et al., 2020b). SARS-CoV-2 is the third coronavirus in the past two decades, responsible for a serious outbreak and health threat (Cui, Li & Shi, 2019; Peeri et al., 2020; Yang et al., 2020; Zhou et al., 2020b; Zhu et al., 2020). The other two outbreaks caused by coronaviruses are now known as Severe Acute Respiratory Syndrome (SARS) and Middle East Respiratory Syndrome (MERS) (Cui, Li & Shi, 2019; Forni et al., 2017; Holmes & Rambaut, 2004). In view of the major global health impact of SARS-CoV-2 or “COVID-19 virus”, which has now infected >240 million people worldwide resulting in >4.9 million deaths (https://coronavirus.jhu.edu/), it is of much importance to understand the evolution of this virus and genetic features which distinguish it from other coronaviruses.

Coronaviruses are a large group of viruses that are a part of the subfamily Coronavirinae (Cui, Li & Shi, 2019; Forni et al., 2017). Based on their phylogenetic branching and genomic structures, the viruses from the subfamily Coronavirinae have been divided into four genera viz. Alphacoronavirus, Betacoronavirus, Gammacoronavirus, and Deltacoronavirus (Cui, Li & Shi, 2019; Forni et al., 2017). Of these four genera, only a few viruses from the genera Alpha- and Beta-coronavirus infect humans and cause respiratory illness (Cui, Li & Shi, 2019; Forni et al., 2017). Members of the Alphacoronavirus lineage (viz. HCoV-NL63, HCoV-229E) cause only mild disease in humans. In contrast, betacoronaviruses are responsible for different coronavirus epidemics viz. SARS-CoV, MERS-CoV and SARS-CoV-2 (Cui, Li & Shi, 2019; Forni et al., 2017; Lu et al., 2020; Peeri et al., 2020; Zhou et al., 2020b). The genus Betacoronavirus is made up of four separate clusters commonly referred to as A, B, C, and D, which are now recognized as distinct subgenera with the names Embecovirus, Sarbecovirus, Merbecovirus and Nobecovirus, respectively (Cui, Li & Shi, 2019; Forni et al., 2017; Lu et al., 2020; Wong et al., 2019; Woo et al., 2010). In addition, another subgenus, Hibecovirus, has also been added recently (Wong et al., 2019). Based on phylogenetic studies, the MERS-CoV groups within the subgenus Merbecovirus, whereas both SARS-CoV and SARS-CoV-2 viruses as well as many other bat-SARS-related (SARSr)-CoVs are part of the subgenus Sarbecovirus (Cui, Li & Shi, 2019; Lu et al., 2020; Wong et al., 2019; Zhou et al., 2020b). Thus, from the viewpoint of understanding the origin and evolution of SARS-CoV-2 virus, it is important to determine how this virus differs from SARS-CoV and other bat-CoVs within the subgenus Sarbecovirus (Andersen et al., 2020; Lau et al., 2020; Li et al., 2020; Lu et al., 2020; Wan et al., 2020; Zhang & Holmes, 2020; Zhou et al., 2020b).

Phylogenetic and amino acid sequence similarity studies indicate that at the whole genome level SARS-CoV-2 is most closely related (96.1% identity) to a bat CoV (Bat-CoV-RaTG13) isolated from a bat Rhinolophus affinis (Andersen et al., 2020; Zhou et al., 2020b), followed by 93.3% identity to a bat-CoV isolate RmYN02 (Zhou et al., 2020a), 92.64% identity to bat-CoV isolate RacCS203 from Thailand (Wacharapluesadee et al., 2021), 92.6% identity with two novel Beta-CoVs (RShSTT182 and RShSTT200) sampled from Cambodia (Hul et al., 2021), 91.02% identity to a pangolin-CoV (strain MP789) (Andersen et al., 2020; Lam et al., 2020; Xiao et al., 2020; Zhang, Wu & Zhang, 2020) and 90.7% identity to a bat-CoV-PrC31 isolated from Yunnan province of China (Li et al. 2021). In contrast, SARS-CoV-2 exhibits lower (80–88%) sequence identity to the SARS-CoV (Andersen et al., 2020; Lu et al., 2020; Zhou et al., 2020b). These observations indicate that SARS-CoV-2 has originated from other bat/pangolin CoVs, independently of the SARS-CoV (Andersen et al., 2020; Zhou et al., 2020b). However, while the genome of SARS-CoV-2 is most similar to the Bat-CoV-RaTG13 virus, the sequence of its receptor binding domain (RBD), which plays a critical role in the binding of this virus with the human angiotensin-converting enzyme 2 (ACE2) receptor (Letko, Marzi & Munster, 2020; Wan et al., 2020), exhibits maximal similarity to the pangolin-CoV_MP789 virus rather than to the Bat-CoV-RaTG13 virus (Andersen et al., 2020; Lam et al., 2020; Lau et al., 2020; Zhang & Holmes, 2020). As genetic recombination is common among RNA viruses including SARS-CoV and SARS-CoV-2 (Cui, Li & Shi, 2019; Lam et al., 2020; Lau et al., 2015; Lau et al., 2020; Li et al., 2020), this raises the possibility that this process may have played an important role in the evolution of SARS-CoV-2 and pangolin-CoV_MP789 viruses.

Most of the studies on understanding the origin of SARS-CoV-2 and evolutionary relationships among coronaviruses/sarbecoviruses are based on phylogenetic tree constructions and sequence similarity determination for different genes/proteins from these viruses (Li et al., 2020; Liu et al., 2020; Lu et al., 2020; Zhou et al., 2020b). In addition, extensive work has been carried out on the epidemiology and evolution of different SARS-CoV-2 strains and also developing methods for identifying and predicting SARS-CoV-2 and other pathogenic viruses (Gupta et al., 2021; Issa et al., 2020; Parlikar et al., 2020) (Saha et al., 2021). While these studies have provided important insights into the evolution of SARS-CoV-2 and related viruses, it will be useful to carry out further studies on understanding the evolution of SARS-CoV-2, and its novel sequence characteristics, by means of other independent approaches. Based on genome sequence data, one approach that has proven very useful for understanding evolutionary relationships consists of identification of conserved signature indels (insertions/deletions) (CSIs) in genes/proteins that are uniquely shared by either a given group of organisms/viruses or commonly shared by different groups/lineages (Baldauf & Palmer, 1993; Gupta, 2014). Due to the rare and discrete nature of genetic changes that give rise to CSIs, their presence or absence in different lineages is generally not affected by factors that limit the reliability of inferences from phylogenetic trees (Gupta, 2016; Rokas & Holland, 2000; Springer et al., 2004). Because of the above characteristics, CSIs have been widely used for understanding the evolutionary relationships and for demarcation of prokaryotic and eukaryotic organisms (Baldauf & Palmer, 1993; Gupta, 2016; Gupta, Lo & Son, 2018; Springer et al., 2004). However, this approach has been used to only a limited extent for understanding the evolutionary relationships amongst viruses (Lau et al., 2015; Zhou et al., 2020a).

We describe here the results of our phylogenetic studies and a CSI-identification based approach on sarbecoviruses genomes to understand the origin and novel molecular features of SARS-CoV-2 and other sarbecoviruses. Our analyses have identified 17 novel CSIs in the spike (S) and nucleocapsid (N) proteins, which serve to clearly demarcate a number of distinct clusters of Sarbecovirus. In addition, our analyses have identified several CSIs in the N- and C-terminal domains of the spike protein S1 subunit that are commonly shared by the SARS-CoV-2 viruses and other sarbecoviruses belonging to either the SARS-CoV/RShSTT clades or another clade of bat coronaviruses (CoVZC/PrC31 clade). The shared presence of these CSIs between two distinct lineages of sarbecoviruses, in conjunction with the results of amino acid sequence similarities studies, provides evidence for two different genetic recombination events leading to the origin of SARS-CoV-2 and related viruses. We also briefly discuss the structural and functional significance of the CSIs that are specific for the SARS-CoV-2 and other clades of sarbecoviruses.

Material and Methods

Construction of phylogenetic trees and sequence similarity analysis

Protein sequences for the spike (S) and RNA-dependent RNA polymerase (RdRp) proteins from representative viral species/isolates from different lineages of Beta-CoVs were retrieved from the NCBI database and GISAID database of SARS CoV-19 sequences (Elbe & Buckland-Merrett, 2017; Shu & McCauley, 2017). As the focus of this study is on the Sarbecovirus subgenus, more sequences were used for this group in comparison to the other lineages of Beta-CoVs. Sequences for three infectious variants of SARS-CoV-2 including SARS-CoV-2/B.1.617.2 (Delta variant), SARS-CoV-2/ B.1.1.7 (Alpha variant) and SARS-CoV-2/ P.1 (Gamma variant) were also included in analysis. Multiple sequence alignments were created using the ClustalW algorithm from the MEGA6 software package (Kumar et al., 2008). After removal of poorly aligned regions using the Gblocks_0.91b program (Talavera & Castresana, 2007), maximum-likelihood phylogenetic trees based on 100 bootstrap replicates of these sequences were created using the JTT matrix-based model employing MEGA6 (Talavera & Castresana, 2007). All positions with less than 95% site coverage were not considered during analysis. Pairwise sequence similarity matrices of protein sequences were determined using the Clustal Omega program (Sievers et al., 2011).

Identification of CSIs in protein sequences

Multiple sequence alignments of the S and N proteins were created as above and examined manually to identify insertions or deletions (indels), which were specifically found in some or all sarbecoviruses and which were flanked by at least 4–5 conserved amino acid residues in the neighbouring 40–50 residues (Gupta, 2014). The indels which were not flanked by conserved regions were not further considered, as they do not provide reliable molecular characteristics (Gupta, 2014; Gupta, 2016). The indels specific for other subgenera of Beta-CoVs were also not investigated in this study. Using the query sequences, encompassing the indels and of their flanking 40–50 amino acids, another BLASTp search was carried out using the NCBI non-redundant database. All significant hits obtained from these BLASTp searches were examined to determine the group specificities of the identified CSIs. Signature files for the CSIs were created using SIG_CREATE and SIG_STYLE programs described in our earlier work (Gupta, 2014). Sequence information in different figures is shown for only a limited number of strains from different groups/lineages of sarbecoviruses. However, unless otherwise stated, the described CSIs are specific for the indicated clusters and they are also present in other CoVs from these clusters.

Homology modelling of the proteins to map the locations of CSIs in protein structures

The structural locations of the identified CSIs were mapped in the spike protein structures using the experimentally solved structures by creating homology models using the available structures as templates (Wrapp et al., 2020; Yan et al., 2020). Homology models for the N-terminal domain for SARS-CoV-2/Wuhan-Hu-1 (accession number YP_009724390) and Bat-SARS-like-CoVZC45 (accession number AVP78031) were created based on the available experimental structure of SARS-CoV-2 spike protein (PDB: 6vsb) (Wrapp et al., 2020). Cyro-EM based structure for the receptor-binding domain (RBD) of SARS-CoV-2 spike protein (PDB: 6m17, Chain E) was utilized to create a homology model of the RBD for BAT SARS-CoVZC45 to map the structural location of the CSI in SARS-CoV-2/Wuhan-Hu-1. Homology modeling was carried out using the MODELLER v9.11 program (Eswar et al., 2007) and their stereochemical properties were assessed as described in our earlier work (Khadka & Gupta, 2017).

Figure 1 A maximum-likelihood distance tree based on sequence alignment of the spike protein from representative viruses/strains of the genus Betacoronavirus.

The tree was constructed as described in the Methods section and the % bootstraps for different branches are indicated on the nodes. The clades corresponding to different subgenera within the Betacoronavirus as well as other clusters and lineages within the subgenus Sarbecovirus are labeled.

Results

Phylogenetic and sequence similarity analysis of betacoronaviruses

We have constructed phylogenetic tree(s) for betacoronaviruses based on the spike and RdRp protein sequences. The tree for the spike protein is shown in Fig. 1. The tree based on RdRp sequences, which generally shows similar relationships as seen in Fig. 1 (Zhang, Wu & Zhang, 2020), is provided in Fig. S1. In both these trees, the Beta-CoVs form four main clusters corresponding to the four subgenera (Cui, Li & Shi, 2019; Forni et al., 2017; Wong et al., 2019). The clusters for the four subgenera are marked in the tree along with their commonly known clade designations (i.e., clades A, B, C, and D). The viruses from these four subgenera are separated from each other by long-branches and supported by 100% bootstrap scores. Of these four lineages, members of the subgenus Sarbecovirus which contains both SARS-CoV, SARS-CoV-2 and related viruses form a tight cluster in the tree (Fig. 1). Based on this tree, it is possible to draw some inferences, which are generally in accordance with earlier studies (Boni et al., 2020; Lau et al., 2020; Liu et al., 2020; Zhang, Wu & Zhang, 2020; Zhou et al., 2020a). These include (i) The SARS-CoV-2 strains group reliably with the Bat-CoV-RaTG13 and pangolin CoVs in a clade that we have designated as the SARS-CoV-2r (related) cluster. Within this clade, Bat-CoV-RaTG13 exhibits a closer relationship to the SARS-CoV-2 than the pangolin CoVs. (ii) Eight bat SARS-like CoVs viz. CoVZXC21, CoVZC45, PrC31, RShSTT182, RShSTT200, RmYN02, RacCS203 and Rc-o319 form the nearest relatives of the SARS-CoV-2r cluster in the tree. The clade for three of these viruses viz. CoVZXC21, CoVZC45, and PrC31, will be referred to as the CoVZC/PrC31 cluster. Of the remaining five closely-related CoVs, two bat SARS-like CoVs from Cambodia (viz. RShSTT182, RShSTT200) form the immediate outgroup of the SARS-CoV-2r clade and this clade will be referred to as the RShSTT cluster. (iii) The SARS and SARS-like CoVs form a separate clade (marked as Sarbecovirus lineage-1) distinct from the clade encompassing SARS-CoV-2r, CoVZC/PrC31, RShSTT clusters and a cluster grouping the other bat-SARS-like CoVs (labeled as Sarbecovirus lineage-2). (iv) Three other CoV strains viz. SARS-like CoV-BtkY72, Bat-CoV/BB9904/BGR/2008 and Bat-CoV-BM48-31/BGR/2008 formed a separate deeper-branching clade in the tree, which we have designated as the Sarbecovirus lineage-3.

Identification and characteristics of conserved signature indels in the spike and nucleocapsid proteins

The main focus of this work is on using the CSI-identification approach to examine the evolutionary relationships among sarbecoviruses. For this purpose we have examined the sequence alignments of S, N and RdRp proteins for the presence of any useful CSIs. In the RdRp protein, no useful CSI specific for members of the subgenus Sarbecovirus was identified. However, within the S- and N- proteins, we have identified several informative CSIs whose descriptions and evolutionary significance are described below.

Figure 2 Partial sequence alignments of three conserved regions from the S1-N-terminal domain showing a number of CSIs that are specific for different clades/lineages of sarbecoviruses.

(A) shows a 6 aa insert (①) in a conserved region that is present in different viruses from the SARS-CoV-2r cluster, except pangolin-CoV_MP789. In the same position, a 3 aa insert (v) is commonly shared by the two CoVs from the CoVZC/PrC31 cluster and pangolin-CoV_MP789. (B) shows a 6 aa insert (③) in the S1-NTD that is specific for the SARS-CoV-2r cluster except pangolin-CoV_MP789. The CoVs from the CoVZC/PrC31 cluster and pangolin-CoV_MP789 contain a 5 aa insert (❹) in this position. The 2 aa CSI present in this region (labeled as ⑤), is specific for the B-1 lineage of Sarbecovirus comprised of SARS-CoV and related viruses. (C) The CSI indicated as ❻marks a region within the S1-NTD where a 7 aa insertion is present in SARS-CoV-2 strains, Bat-CoV RaTG13 and one of the pangolin strain (GX-P2V). Most pangolin homologs have a 5 aa insert (highlighted in blue and marked A in this position, whereas CoVs from the CoVZC/PrC31 cluster and pangolin-CoV_MP789 contain a 6 aa insert (highlighted in purple and marked B in this position). Dashes (-) in these alignments denote identity with the amino acid shown in the top sequence. The numbers on the top indicate the position within the indicated proteins.

Sequence characteristics of the CSIs in the N-terminal domain of the Spike protein

Based on phylogenetic branching (Fig. 1), SARS-CoV-2 is most closely related to Bat-CoV-RaTG13 and pangolin CoVs, forming the SARS-CoV-2r cluster. We have identified several CSIs which support a close relationship of these viruses and also provide useful insights concerning their evolutionary relationships. In Fig. 2, we show partial sequence alignments of three conserved regions from the S1-N-terminal domain (NTD) where CSIs of specific lengths are found in a number of different lineages of sarbecoviruses. In Fig. 2A, a 6 aa insert in the S1-NTD (boxed and labeled ①) is specifically present in all members of the SARS-CoV-2r cluster with the exception of pangolin-CoV_MP789. Interestingly, the pangolin-CoV_MP789 instead of containing the 6 aa insert specific for the SARS-CoV-2r cluster contained in the same position a 3 aa insert (marked ②), which is commonly shared by the three viruses from the CoVZC/PrC31 cluster. In the sequence alignment for a different region of the S1-NTD shown in Fig. 2B, another 6 aa insert (marked ③) is present, which is also commonly shared by all viruses from the SARS-CoV-2r cluster except pangolin-CoV_MP789. The pangolin-CoV_MP789 contains in this position a shorter 5 aa insert labeled as ❹, which is again commonly shared by the three CoVs from the CoVZC/PrC31 cluster. Importantly, the sequence of this 5 aa insert in the CoVZC/PrC31 strains and pangolin-CoV_MP789 is identical to that found in the SARS-CoV-2r cluster, except that it is shorter by 1 aa residue. The distribution patterns of these two CSIs (i.e., ②and ❹) in different CoVs strains strongly suggests that the pangolin-CoV_MP789 strain is more closely related to the CoVZC/PrC31 cluster of CoVs in the S1-NTD, than to the other pangolin CoVs or viruses from the SARS-CoV-2r cluster (Fig. 1). In addition to the CSIs ③and ❹, the sequence region shown in Fig. 2B also contains a 2 aa CSI (labeled as ⑤), which is specific for the sarbecoviruses comprised of SARS-CoV and related viruses.

Figure 2C shows sequence alignment for another region from the S1-NTD where a 7 aa insertion in a conserved region (labeled ❻) is commonly shared by different SARS-CoV-2 strains, Bat-CoV-RaTG13 and one of the pangolin CoVs (strain GX-P2V). However, all other pangolin GX-strains contain a 5 aa insert in this position and their sequences are almost identical. As the sequence for the strain GX-P2V is very similar to other pangolin CoV_GX strains, but it contains some ambiguity as indicated by a residue marked X, the presence of a 7 aa insert in this strain needs to be verified to exclude the possibility of sequencing error. Nonetheless, as all other pangolin CoV-GX strains contain a shorter 5 aa insert in this position (marked A), this CSI supports a closer relationship of the SARS-CoV-2 to the Bat-CoV-RaTG13 in comparison to the pangolin CoVs. Interestingly, the three bat CoVs from the CoVZC/PrC31 cluster as well as the pangolin-CoV_MP789 contain a 6 aa insert in this position (marked B) again supporting the inference from the CSIs ②and ❹that the pangolin-CoV_MP789 in its S1-NTD is specifically related to the CoVZC/PrC31 cluster of CoVs. In addition, two CoV strains from Cambodia viz. RShSTT182 and RShSTT200 contain a 4 aa insert in this position (marked C), distinguishing them from others. This sequence region also contains 6–7 aa insertions in the CoVs from Sarbecovirus lineages 2 and 3, but their insert sequences are divergent. However, members of the Sarbecovirus lineage-1, which consists of SARS-CoV and related viruses, do not contain an insertion in this position.

Figure 3 Partial sequence alignments of three conserved regions of the spike protein showing a number of CSIs that are specific for different lineages.

The two CSIs shown in (A) (⑦) and (B) ❽are from an (overlapping) region encompassing the receptor binding domain (RBD) of the spike protein. These CSIs are mainly found in the SARS and SARS-CoV-2r clusters of viruses. The CSI ❽is also present in two bat-Cambodia-viruses from the RShSTT cluster. The sequence region comprising the RBD is shown in dark blue, whereas the overlapping parts of the sequence are marked in light blue. (C) shows a 4 aa insertion (❾) present at the S1–S2 subunit junction that is only found in the SARS-CoV-2 strains. In addition, a 1 aa insertion (❿) is also present in this position, which is specific for the sarbecovirus lineage-3. Dashes (-) in the sequence alignments denote identity with the amino acid shown in the top line.

Sequence characteristics of the CSIs in the C-terminal domain of the Spike protein

The S1-CT also contains two CSIs and information for these is shown in Figs. 3A and 3B. These insertions have been described previously (Boni et al., 2020; Lau et al., 2015; Zhang, Wu & Zhang, 2020; Zhou et al., 2020a), however, their evolutionary significance was not clear until recently (Lau et al., 2020; Li et al., 2020). Both these CSIs are present in close proximity within the receptor-binding domain (RBD) of the spike protein (sequence of the RBD is shown in magenta with overlapping sequence (residues 450–469) underlined). Figure 3A shows a 5 aa insert (marked ⑦) that is commonly shared by members of the SARS-CoV-2r clade as well as different SARS and SARS-like CoVs that are part of the Sarbecovirus lineage-1 (see Fig. 1). However, this CSI is not found in other sarbecoviruses including members of the CoVZC/PrC31 cluster, or other bat-SARS-related CoVs such as RShSTT182, RShSTT200. Figure 3B shows another CSI, in this case a 13 aa insertion (marked ❽) which is also commonly shared by members of the SARS-CoV-2r clade as well as different SARS and SARS-like CoVs of Sarbecovirus lineage-1. In addition, a similar CSI is also present in the two strains from RShSTT cluster and Sarbecovirus lineage-3, but it is lacking in members of the CoVZC/PrC31 cluster and in Sarbecovirus lineages-2. It should also be noted that both these CSIs are present in all of the pangolin homologs including the pangolin CoV_MP789. Furthermore, it is also of much interest that the amino acid sequence of pangolin-CoV_MP789 in the RBD region is highly similar to the SARS-CoV-2, but it differs significantly from the sequences of CoVs from the CoVZC/PrC31 cluster. Figure 3C shows a 4 aa insert (marked ❾), which is only present in SARS-CoV-2 and its variants, but it is not found in any virus from the SARS-CoV-2r cluster or any other Sarbecovirus. This insert, which is located at the junction of the S1/S2 subunit creating a furin-cleavage motif, has been described previously (Andersen et al., 2020; Jaimes et al., 2020; Zhang, Wu & Zhang, 2020; Zhou et al., 2020a). In the same position where this 4 aa insert is found, a 1 aa insertion (marked ❿) is also present in members of the Sarbecovirus lineages-3 and one other CoV.

Table 1 Conserved indels identified in the spike and nucleocapsid proteins that are specific for various groups, lineages and sub-lineages of Betacoronavirus.

Protein name	Accession number	Indel length	Indel location	Indel specificity	Figure No:	
Spike	YP_009724390	6 aa ins	229–285	SARS-CoV-2r cluster	Fig. 2A	
Spike	AVP78042	3 aa ins	229–285	Bat-SARSr-CoVZC/PrC31 cluster	Fig. 2A	
Spike	YP_009724390	6 aa ins	134–187	SARS-CoV-2r cluster	Fig. 2B	
Spike	AVP78042	5 aa ins	134–187	Bat-SARSr-CoVZC/PrC31 cluster	Fig. 2B	
Spike	ACZ72108	2 aa ins	134–187	Sarbecovirus Lineage-1SARS and SARS-relatedCoVs	Fig. 2B	
Spike	YP_009724390	5–7 aa ins	51–96	SARS-CoV-2r cluster	Fig. 2C	
Spike	YP_009724390	5 aa ins	419–469	SARS-CoV-2r cluster	Fig. 3A	
Spike	YP_009724390	13 aa ins	450–517	SARS-CoV-2r cluster	Fig. 3B	
Spike	YP_009724390	4 aa ins	662–700	SARS-CoV-2r cluster	Fig. 3C	
Spike	YP_003858584	1 aa ins	662–700	Sarbecovirus Lineage-3	Fig. 3C	
Nucleocapsid	YP_009724397	2 aa del	385–417	SARS-CoV-2r cluster & Bat-SARSr-CoVZC/PrC31 cluster & Other Bat-SARS CoV-2 related CoV strains	Fig. S1A	
Spike	APO40579	1 aa del	157–191	Sarbecovirus Lineage-3	Fig. S1A	
Spike	APO40579	1 aa ins	361–392	Sarbecovirus Lineage-3	Fig. S2B	
Spike	APO40579	2 aa ins	601–636	Sarbecovirus Lineage-3	Fig. S2C	
Nucleocapsid	YP_003858591	1 aa del	385–417	Sarbecovirus Lineage-3	Fig. S3A	
Nucleocapsid	QIA48648	2 aa del	1–34	Pangolin CoVs	Fig. S3B	
Nucleocapsid	YP_003858591	1 aa del	1–34	Sarbecovirus Lineage-3	Fig. S3B	

CSIs in S- and N- Proteins specific for other Sarbecovirus lineages

We have also identified a number of other CSIs in the S- and N- proteins that are uniquely shared by other lineages of Sarbecovirus. Sequence alignments for these CSIs are provided in Figs. S2 and S3 and some information regarding the characteristics of these CSIs is summarized in Table 1. Of these CSIs, three CSIs in the S-protein (labeled as ①, ②and ③) as well as two CSIs in the nucleocapsid protein (marked ⑤and ⑦) are specific for the Sarbecovirus lineages-3 (Figs. S2 and S3), indicating this to be a distinct, deep-branching, lineage of Sarbecovirus. Of the other two CSIs found in the N-protein, the CSI labeled ④in Fig. S3A) is commonly shared by all members of the SARS-CoV-2r cluster as well as by all other bat- and pangolin- CoVs closely related to SARS-CoV-2 including members of the CoVZC/PrC31 cluster. In addition, the N-protein also contains another CSI (labeled ⑥in Fig. S3B) which is commonly shared by all pangolin-homologs except pangolin-CoV_MP789.

Sequence similarity studies on SARS-CoV-2 and related viruses/ strains providing evidence of recombination

The identified CSIs in the spike protein show contradictory results (elaborated in the Discussion section) for two sets of CoVs, specifically (i) the pangolin CoV-strain_MP789 and (ii) the SARS-CoV-2r cluster of viruses. To investigate this further, we have determined the pairwise amino acid identity of the spike protein from SARS-CoV-2 and pangolin-CoV_MP789 to representative CoVs from the CoVZC/PrC31, SARS and SARS-related lineage, and other clusters. These determinations were made for the entire spike protein as well as its specific sequence regions including the S1-NTD, S1-CTD, and the C-terminal region containing S2 subunit. Results from these analyses are presented in Fig. 4.

Figure 4 Amino acid sequence identity matrix showing the percentage (%) of sequence identity for different regions of the Spike protein from representative viruses/strains of the subgenus Sarbecovirus.

Individual cells in the matrix are colored shaded in gradients ranging from blue at the lowest (40%) to red at the highest (100%).

As seen from Fig. 4, for the full length spike protein, highest sequence similarity of pangolin-CoV_MP789 is observed for SARS-CoV-2 (90.51%). This value is lower than the similarity of SARS-CoV-2 to Bat-CoV-RaTG13 (97.71%) as well as other pangolin CoV strains such as GX-P2V (92.38%). However, for the S1-NTD region (aa 1–307), the pangolin-CoV_MP789 is most similar to the Bat-SARS-like-CoV-CoVZC45 (85.81%), whereas all other CoVs including SARS-CoV-2, Bat-CoV-RaTG13, SARS CoV-ExoN1 and other pangolin CoVs-GX strains exhibited much lower similarity (45–67%). A more dramatic difference in sequence similarity of pangolin-CoV_MP789 was observed when sequence comparison was made for the S1-CTD region (aa 319–540) or for the C-terminal region containing the S2-subunit (i.e., from aa 541–1265). For these two sequence regions, pangolin-CoV_MP789 exhibited 96.86% and 98.49% sequence identity, respectively, to the SARS-CoV-2 virus and these values were higher than that seen for any other CoV including Bat-CoV-RaTG13. We have also examined the multiple sequence alignment of the spike protein from pangolin-CoV_MP789 and representative CoVs from SARS-CoV-2r and CoVZC/PrC31 clusters for polymorphic sites where similar amino acids are present in different CoVs (Fig. S4 and Fig. S5). These analyses show that within the S1-NTD (aa 1–320), in >80% of the polymorphic sites, pangolin-CoV_MP789 and the CoVZC/PrC31 cluster of CoVs contain the same amino acids. In contrast, for the remainder of the S-protein (aa 330–1265), in >90% of the polymorphic sites, identical amino acids are present in pangolin-CoV_MP789 and SARS-CoV-2 CoVs. We have also examined the nucleotide sequence of the spike protein from pangolin-CoV_MP789 and other CoVs for any interesting feature that might be present near the location (near aa 320–330), where a shift in sequence similarity occurs from the CoVZC/PrC31-like sequence to the SARS-CoV-2 like sequence. Interestingly, as shown in Fig. S6, in the nucleotide sequence of pangolin-CoV_MP789 and other closely related viruses, a restriction site for the enzyme Ssp1 is present (marked with arrow, makes a blunt-end cut), at the position where transition from one sequence type to the other occurs. The significance of these observations is discussed in the Discussion section.

The S1-CTD of SARS-CoV-2 contains two CSIs (⑦and ❽), which besides the SARS-CoV-2r cluster of CoVs (including pangolin-CoV_MP789) are only found in the Bat-SARS-like-CoVs (Sarbecovirus-lineage 2). To understand the significance of this observation, we have also determined the sequence similarity of SARS-CoV-2 for representative viruses from SARS-CoV-2r sister clades as well as for the SARS-Exo-N1 virus for different regions of the spike protein (Fig. 4). As seen from Fig. 4, the S1-CTD as well as the remainder of the C-terminal region of the SARS-CoV-2 spike protein, outside of the SARS-CoV-2r cluster, shows maximal sequence similarity (85.32% and 97.66%, respectively) to the corresponding regions of the RShSTT-CoVs (Cambodia CoV strains). In comparison, the sequence similarity of SARS-CoV-2 for the same regions of the SARS-CoV-Exo-N1 virus was much lower (73.42% and 87.77%, respectively).

Localizations of the CSIs in protein structures

We have also mapped the locations of the identified CSIs in the three-dimensional structure of the spike protein. For these studies, using the available structures for the spike proteins (PDB IDs: 6acc, 6m17, 6vsb), homology models were created from a number of different CoVs including CoVZC-45, RShSTT182, SARS ExoN1 and SARS-CoV-2 (Wuhan-Hu-1) to localize and visualize the locations of different identified CSIs (Fig. 5 and Fig. S7).

Figure 5 Mapping the surface locations of some of the identified CSIs in the spike protein by homology modeling.

(A) Homology models of the spike protein from BatSARS-like-CoVZC45 based on the available structures of SARS-CoV-2 (PDB IDs: 6vsb, and 6m17) and SARS-CoV-2 spike proteins (PDB ID: 6vsb). (B) A cartoon and surface representation of a Cryo-EM structure of the SARS CoV spike protein (PDB ID: 6acc). (C) A cartoon and surface representation of a spike protein from SARS-CoV-2. The S1-NTD and S1-CTD domains are homology models based on the experimental structure of the SARS-CoV-2 spike protein (PDB ID: 6vsb, and 6m17). In all of these models, the S1-NTD and S1-CTD domains are shown in cyan and dull green color, respectively, whereas the S2 subunit is shown in bright green color. The identified CSIs in the S1-NTD and S2-CTD domains of these CoVs are labeled and shown in blue and red colors, respectively. The asterisk (*) in (B) indicates that the CSI ⑦is only found in SARS-viruses but not present in bat-RShSTT-CoVs.

Fig. 5A shows a homology model for the spike protein from Bat SARS-like CoVZC45 virus. The three CSIs that are present in the N-terminal domain of S1 are shown in blue and all of them are located in surface-exposed loops/patches on the S1-NTD of the protein. In the CoVZC45 virus, no CSI was identified in the C-terminal domain of S1. In Figure 5B we show a homology model for the SARS-CoV ExoN1 protein. This protein contains two CSIs (5 aa and 13 aa) in the S1-CTD and a small 2 aa CSI in the S1-NTD. Similar to the localization of CSIs in the CoVZC45 virus, all of the CSIs in SARS-CoV ExoN1 are also present in surface-exposed loops on the protein. Figure 5C shows a homology model for the spike protein from a virus (Wuhan-Hu-1) from the SARS-Cov-2r cluster. The structural characteristics of the viruses from the SARS-Cov-2r shows that these viruses contain all of the sequence features (CSIs) that are similar to the S1-NTD of the CoVZC/PrC31 cluster of viruses as well those that are distinctive of the S1-CTD of the SARS viruses. We have also mapped the structural locations of the CSIs in the spike protein that are specific for the CoVs from the Sarbecovirus-lineage-3. The results from these studies are shown in Fig. S7 and all of these CSIs are also located in surface-exposed loops of the spike protein.

Discussion

In this work, we have examined the evolutionary relationships among sarbecoviruses using a sequence-based approach involving identification of conserved inserts or deletions in conserved regions (i.e., CSIs) in protein sequence alignments that are uniquely shared by either a specific group of Sarbecovirus, or are commonly shared by specific lineages of sarbecoviruses. Results presented here have identified >15 CSIs in the S- and N-proteins that reliably demarcate a number of clades of sarbecoviruses and also provide novel insights into the evolution of SARS-CoV-2r cluster of viruses and a pangolin-CoV (viz. pangolin-CoV_MP789), which in its S1-CTD region of the spike protein is most closely related to the SARS-CoV-2 virus. To understand the evolutionary significance of the identified CSIs, we have also carried out sequence similarity studies on different regions of the spike protein. Some characteristics of the identified CSIs and the inferences that can be drawn from them, in conjunction with the results from other analyses presented here, are discussed below.

Of the CSIs identified, two CSIs (①and ③) in the S1-NTD are specific for most CoVs from the SARS-CoV-2r cluster (viz. SARS-CoV-2 and its variants, Bat-CoV-RaTG13 and all other pangolin CoVs except pangolin-CoV_MP789). Another 7 aa long CSI in the S1-NTD (❻) is only present in SARS-CoV-2 and its variants and Bat-Cov-RaTG13. As the CoVs sharing these CSIs form a monophyletic grouping in the tree, the most parsimonious explanation to account for the shared presence of these CSIs is that the genetic changes responsible for these CSIs occurred in a common ancestor of the SARS-CoV-2r lineage. These CSIs provide strong evidence independent of the phylogenetic trees, that the SARS-CoV-2 virus is closely related to the other viruses in the SARS-CoV-2r cluster. Within the S1-NTD sequence, three smaller CSIs, related to those found in the SARS-CoV-2r cluster, are also present in the three CoVs from the CoVZC/PrC31 cluster as well as in pangolin-CoV_MP789. These results indicate that in the S1-NTD region, pangolin-CoV_MP789 shows a close relationship to the CoVs from the CoVZC/PrC31 cluster. Two other large and informative CSIs are present in the S1-CTD (Fig. 3, ⑦and ❽) of the spike protein. In this case, CSIs of similar lengths are present in all of the viruses from the SARS-CoV-2r cluster (including pangolin-CoV-MP789) and by the Sarbecovirus lineage-1 consisting of the SARS and SARS-related viruses. However, both these CSIs are absent in the CoVZC/PrC31 cluster of CoVs. Of these two CSIs, the larger 12-13 CSI ❽is also present in the two viruses from the RShSTT cluster. The distribution patterns of various identified CSIs in the spike protein show contradictory results for two groups of CoVs i.e., pangolin-CoV-MP789 and SARS-CoV-2r cluster of viruses. In Fig. 6, we summarize the nature of these anomalous observations and our suggested explanations (models) for these observations and their implications regarding the origin of the spike protein from these CoVs.

Figure 6 A conceptual diagram (models) summarizing the sequence characteristics of the spike protein from specific lineages of sarbecoviruses and their implications regarding the origin of Pangolin-CoV_MP789 virus and SARS-CoV-2r cluster of viruses.

(A) Sequence characteristics of the S1-NTD and S1-CTD from SARS-CoVZC45, a SARS-CoV-2r virus and pangolin-CoV_MP789 and a model based upon them for the origin of the pangolin-CoV_MP789 virus. The ? mark besides the SARS-CoV-2r virus indicates that the virus involved in this recombination remains unidentified. The percent identity (96.86%) shown here is for the Wuhan-Hu-1 virus. The CSIs that are present (or predicted to be present) in these sequence regions and their sequence lengths are noted above the lines. The % amino acid sequence identity noted on the top line is to the indicated region to the pangolin-CoV_MP789 virus. The cross (x) in (A) and (B) indicates a genetic recombination event, which is postulated to have occurred at (near) the S1-NTD and S1-CTD boundary, where a Ssp1 site (marked by *) is present (B) The sequence characteristics and distribution pattern of different CSIs in the S1-NTD and S1-CTD of SARS-CoVZC45, RShSTT182/200 and SARS-CoV-2r cluster of CoVs and a model based on them for the origin of the SARS-CoV-2r cluster of viruses. The numbers below the lines in parenthesis indicate the % amino acid sequence identity of the indicated regions to the SARS-CoV-2 Wuhan-Hu-1 virus. The intermediate is a postulated stage resulting from recombination, prior to the occurrence of further evolutionary changes in it. The vertical arrows indicate other genetic changes in the evolution of SARS-CoV-2r virus including acquisition of the 4 aa insertion (❾) by SARS-CoV-2 at the S1 and S2 boundary. The * indicates the presence of a Ssp1 restriction site in the sequences of these viruses, which is indicated to be the presumed site of recombination.

The first set of anomalous observations concerns the pangolin-CoV_MP789, which branches with the SARS-CoV-2r cluster in phylogenetic trees (see Fig. 1). For all of the CSIs present in the S1-NTD (viz. ①, ②, ③, ❹, ❻and ❻A), the sequence characteristics of this virus are identical to those present in the CoVZC/PrC31 cluster of CoVs and they differ from those seen for other members of the SARS-CoV-2r cluster. However, in the S1-CTD region, the pangolin-CoV_MP789 shares two CSIs (⑦and ❽), which are shared characteristics of the SARS-CoVs and the SARS-CoV-2r cluster of CoVs, but which are absent in the CoVZC/PrC31 cluster of viruses. The anomalous nature of the spike protein sequence from pangolin-CoV_MP789 is also readily apparent from the results of amino acid sequence similarity studies on different regions of the protein. The S1-NTD region (1–307 aa) of pangolin CoV_MP789 shows highest similarity (85.81%) to the CoVZC/PrC31 cluster of viruses, while its S1-CTD domain is highly similar (96.86%) to the SARS-CoV-2 virus and shows much lower similarity (68.34%) to the CoVZC45. The simplest and most plausible explanation to account for the distribution patterns of the identified CSIs, and the results of sequence similarity studies, is that the spike protein from pangolin-CoV_MP789 has originated by recombination between the S1-NTD region of a virus from the CoVZC/PrC31 cluster and the C-terminal region of a virus that is closely related to SARS-CoV-2 (Fig. 6A) (Gupta & Khadka, 2020). Following the recombination event, subsequent genetic changes were likely responsible for the differences in the lengths of the CSIs between CoVZC/PrC31 and SARS-Cov-2r clade (Gupta & Khadka, 2020). Recently, the origin of pangolin-CoV_MP789 by a recombination event has also been suggested by others (Lau et al., 2020; Liu et al., 2020). Based on sequence alignment of pangolin-CoV_MP789, SARS-CoV-2, and CoVZC/PrC31 cluster of viruses (Fig. S4), it is possible to pinpoint the approximate breakpoint for this recombination event. In this sequence alignment, the N-terminal 1–320 aa of pangolin-CoV_MP789 matches closely with the CoVZC45 CoV, whereas the rest of its sequence (from aa 330–1265) is most similar to the SARS-CoV-2 virus. Thus, the recombination break-point within the spike protein is indicated to lie between aa 320–330 in the SARS-CoV-2 sequence. Interestingly, this sequence region of the spike protein contains a restriction site for the enzyme Ssp1 (Fig. S6), which likely constitutes the presumed recombination site.

The pangolin-CoV_MP789 is also of much interest for understanding the origin of SARS-CoV-2. Of all the CoVs that have thus far been identified, the Bat-CoV-RaTG13 is considered to be most closely related to the SARS-CoV-2. However, the sequence of pangolin-CoV_MP789 beyond the recombination breakpoint i.e., for the S1-CTD region (aa 319–540, which includes the RBD) shows much higher sequence similarity to the SARS-CoV-2 (96.86%) in comparison to the Bat-CoV-RaTG13 (90.13%). In the receptor binding region, the sequence of this virus differs from SARS-CoV-2 in only 1 amino acid, whereas the corresponding sequence from Bat-CoV-RaTG13 has 17 different aa substitutions (Andersen et al., 2020; Liu et al., 2020; Zhang, Wu & Zhang, 2020). The remainder of the C-terminal sequence (i.e., aa 541–1265) of pangolin-CoV_MP789 is also highly similar (98.5% sequence identity) to the SARS-CoV-2 virus. Currently, the sequence of the S1-NTD region of the virus that has given rise to pangolin-CoV_MP789 is not known. However, our results suggest that the pangolin/bat species may harbor another CoV, whose both S1-NTD and S1-CTD regions are more similar to the human SARS-CoV-2 than all other known CoVs including the Bat-CoV-RaTG13. This putative virus could represent the immediate or closest ancestor of the SARS-CoV-2. In view of the unusual sequence characteristics of the pangolin-CoV_MP789, it is of interest whether the recombination event leading to this virus occurred in a bat/pangolin species, or could this virus sequence be chimeric resulting from an inadvertent combination of sequences from two different viruses during metagenomic assembly (Gupta & Khadka, 2020; Liu et al., 2020).

The second set of anomalous observations concerns the sequence characteristics of the spike protein from SARS-CoV-2r CoVs (Fig. 6B). As noted above, in the S1-NTD region, both SARS-CoV-2r and CoVZC/PrC31 cluster of CoVs contain a number of CSIs in the same position that are related in sequence, but which are not found in the SARS and SARS-related viruses. Although the presence of similar CSIs in the same position can also result from convergent evolution, the fact that the S1-NTD region of the SARS-CoV-2r viruses also exhibits maximal sequence similarity (66.77%) to the CoVZC/PrC31 cluster of CoVs strongly suggests that the S1-NTD region of the SARS-CoV-2r viruses, where these CSIs are found, is derived from CoVZC/PrC31 cluster of CoVs. In contrast to these observations, the S1-CTD domain of SARS-CoV-2r contain two CSIs (⑦and ❽), which besides the SARS-CoV-2r CoVs are mainly found in all SARS and SARS-like CoVs (Gupta & Khadka, 2020; Lau et al., 2015; Lau et al., 2020; Li et al., 2020). One of these CSIs (CSI ❽) is also present in the two CoVs from the RShSTT cluster. However, both these CSIs are not found in the CoVZC/PrC31 cluster of CoVs. Furthermore, of all the viruses that lie outside of the SARS-CoV-2r cluster, the RShSTT virus followed by SARS virus (ExoN1) are most similar to the SARS-CoV-2 (85.32% 76.13% identity, respectively) in their sequences for the S1-CTD region (see Fig. 4). The simplest explanation to account for these contrasting sequence characteristics of the SARS-CoV-2r viruses is that this cluster of virus has originated by a recombination between the S1-NTD region from a CoVZC/PrC31 cluster of CoV and the C-terminal region (including the S1-CTD) of a virus related to the RShSTT cluster (or a SARS-like) of virus.

It should be noted that while the SARS- and SARS-like viruses contain both CSIs ⑦and ❽, the RShSTT cluster of CoVs contains only the CSI ❽. Thus, it is likely that the CSI ⑦was introduced in the common ancestor of SARS-CoV-2r clade by either a secondary recombination event involving a SARS-related virus or independently by means of convergent evolution due to its selective advantage (Gupta, 2018; Khadka, Persaud & Gupta, 2019). Based upon available information, we are unable to distinguish between these two possibilities. Sequence comparisons of the SARS-CoV-2, CoVZC/PrC31, and RShSTT cluster of CoVs (Fig. S5), also indicate that this recombination event also occurred close to the S1-NTD and S1-CTD boundary, where the Ssp1 restriction site is present. Lastly, the SARS-CoV-2 virus contains a distinctive 4 aa insert (❾) at the junction of S1 and S2 subunit (Andersen et al., 2020; Jaimes et al., 2020; Zhang, Wu & Zhang, 2020; Zhou et al., 2020a). This insertion creates a furin-cleavage motif that enhances the host cell entry/membrane fusion of the virus (Belouzard, Chu & Whittaker, 2009). However, unlike other identified CSIs, which are commonly shared by a number of bats and pangolin sarbecoviruses, this 4 aa gain of function insertion is found only in the SARS-CoV-2 strains, leading to a controversy concerning its origin (Coutard et al., 2020; Lundstrom et al., 2020; Segreto & Deigin, 2021).

The spike protein plays a pivotal role in the functioning of CoVs by enabling binding of the virus to its cellular receptors and subsequent fusion with host cell membrane (Grove & Marsh, 2011; Shang et al., 2020; Wan et al., 2020; Yan et al., 2020). In the present work, we have identified many CSIs in the spike protein. Earlier work on CSIs provides evidence that the genetic changes represented by them are functionally important (Alnajar, Khadka & Gupta, 2017; Khadka & Gupta, 2017; Singh & Gupta, 2009). Our analysis shows that all of the identified CSIs in the spike protein are present in surface-exposed loops, which are predicted to play important roles in mediating novel protein-protein interactions (Akiva, Itzhaki & Margalit, 2008; Hashimoto & Panchenko, 2010; Khadka & Gupta, 2017; Singh & Gupta, 2009). It is important to note that the CSIs ⑦and ❽, which are commonly shared by both SARS and SARS-CoV-2 viruses, are located within the receptor binding domain (RBD) of the S-protein (residues 423-494) and they form a significant portion of the receptor binding motif (Gupta & Khadka, 2020; Liu et al., 2020; Lu et al., 2020; Shang et al., 2020; Wan et al., 2020; Zheng, 2020). Earlier studies indicate that these CSIs or residues from them play a critical role in the binding of spike protein to the human ACE2 receptor and viruses lacking these two CSIs either show reduced binding to the ACE2 receptor or do not infect human cells (Lau et al., 2020; Letko, Marzi & Munster, 2020; Li et al., 2020; Shang et al., 2020; Wan et al., 2020; Zhou et al., 2020a). In this context, our identification of three novel CSIs in the N-terminal domain of the spike protein which are specific for the SARS-CoV-2r cluster of viruses is of much interest. These CSIs also form novel surface-exposed loops in the S1-NTD domain of the S-protein (Fig. 5). Based on earlier work, both the S1-NTD as well as the S1-CTD domains of CoVs are involved in binding of the spike protein to different host receptors (Cui, Li & Shi, 2019; Li, 2016). Thus, it is possible that the novel surface-exposed loops/patches formed by these three SARS-CoV-2r-specific CSIs may be playing important roles by enabling specific interactions with other surface-exposed proteins/components in the host cell. These interactions could enhance the binding/entry of viruses in human cells and could contribute towards greater transmissibility and virulence of the SARS-CoV-2 virus (Gussow et al., 2020). Thus, it is of importance to test/confirm the functional roles of these CSIs by experimental studies.

Our analysis has also identified multiple CSIs in the S- and N- proteins that are uniquely found in the Sarbecovirus lineage-3. The members of this lineage also contain the 12–13 aa insertion (❽) (Fig. 3B), which forms a part of the RBD and is involved in making contact with the ACE2 receptor. Hence, it should be of interest to investigate the members of this lineage for their ability to bind to the ACE2 receptor and their infectivity to human cells.

Conclusions

This work reports the identification of many novel CSIs in the spike protein that are specific for different lineages of sarbecoviruses. The distribution patterns of these CSIs in different clades of sarbecoviruses in conjunction with sequence similarity studies and phylogenetic analysis provide important insights into the evolution of SARS-CoV-2 and related viruses. The results presented have identified two important recombination events. One of these events involving recombination between the CoVZC/PrC31 clade and the RShSTT cluster of CoVs is postulated to have formed the common ancestor of SARS-CoV-2r viruses. The second recombination event identified has occurred between the CoVZC45 virus and an unidentified CoV related to SARS-CoV-2. This recombination has resulted in the formation of the Pangolin-CoV-MP_789 virus. The CoV involved in this latter recombination event is postulated to be more closely related to the SARS-CoV-2 than the Bat-CoV-RaTG13 virus, which is currently considered to be most closely related to the SARS-CoV-2. The CSIs specific for different clades of sarbecoviruses, due to their predicted functional importance, also provide novel means for functional studies and other investigations on these viruses.

Supplemental Information

Supplemental Information 1 A maximum likelihood tree based on RNA dependent RNA polymerase protein sequences showing the branching of different coronaviruses

Click here for additional data file.

Supplemental Information 2 Excerpts from the sequence alignment for three regions of the spike protein depicting three CSIs which are specific for the CoVs from Sarbecovirus lineage-3

Click here for additional data file.

Supplemental Information 3 Excerpts from the sequence alignment for nucleocapsid protein depicting a number of CSIs which are specific for Sarbecovirus lineages

(A) shows a 2 aa deletion (④) in the N-protein that is specific for the SARS-CoV-2r, CoVZC/PrC31 clusters as well as other Bat SARS-CoV-2 related strains of CoVs. The viruses from Sarbecovirus lineage-3 contain a separate 1 aa deletion (⑤) in this position. The (B) shows the sequence alignment of N-protein depicting a 2 aa deletion (⑥) that is specific for pangolin CoVs. A 1 aa deletion (⑦) specific for Sarbecovirus lineages-3 is also present in a neighboring region.

Click here for additional data file.

Supplemental Information 4 A multiple sequence alignment of the spike protein from pangolin-CoV_MP789 and representative CoV strains from SARS-CoV-2 and CoVZC/Prc31 clusters showing the chimeric nature of MP789 sequence

In this alignment, the polymorphic positions where the sequence of pangolin CoV_MP789 is identical to that of CoVZC/PrC31 cluster of viruses are highlighted in yellow, whereas the polymorphic positions where the sequence of pangol is similar to SARS-CoV-2r viruses are highlighted in blue. Location containing Ssp1 restriction site is highlighted with red box and labelled.

Click here for additional data file.

Supplemental Information 5 A multiple sequence alignment of the spike protein from SARS coronavirus ExoN1, SARS coronavirus ShanghaiQXC2 and representative CoV strains from SARS-CoV-2r, RShSTT182/200 and CoVZC/PrC31 clusters showing the chimeric nature of SARS-CoV-2 sequence

In this alignment, the polymorphic positions where the sequences of SARS-CoV-2r viruses are identical to that of CoVZC/PrC31 cluster of viruses are highlighted in yellow, whereas the polymorphic positions where the sequence of RShSTT182/200 and SARS coronavirus ExoN1 viruses are similar to SARS-CoV-2r viruses are highlighted in blue. Location containing Ssp1 restriction site is highlighted with red box and labelled.

Click here for additional data file.

Supplemental Information 6 Nucleotide and amino acid sequence of the spike protein from pangolin-CoV_MP789 surrounding the site where sequence changes from CoVZC-like to SARS-CoV-2 like sequence

This Spp1 restriction site is also present (highlighted) in the sequences of other Sarbecoviruses and it is indicated to be the breakpoint in the two recombination events described in this work.

Click here for additional data file.

Supplemental Information 7 Homology model of N-terminal domain (NTD) of spike protein from SARS-related coronavirus BtKY72 (Acc no: APO40579) based on available experimental structure of SARS coronavirus BJ012 spike protein (PDB: 5x58)

The location of identified 1 aa deletion is highlighted. Detailed species distribution information for the 1 aa deletion in Fig. S2A). (B) Homology model of RBD of SARS-related coronavirus BtKY72 spike protein (Acc no: APO40579) based on available experimental structure of SARS coronavirus BJ012 spike protein (PDB: 5x58). The receptor binding domain (RBD) domain is highlighted as green and extended C-terminal region as pale green. The locations of identified signature indels are highlighted red and labelled. Detailed species distribution information for the 1 aa insert and 2 aa insert in RBD in Fig. S2B) and Fig. S2C.

Click here for additional data file.

We thank Drs. Herb Schellhorn and Anjalee Gupta for their reading of the manuscript and many helpful comments to enhance the clarity of the presented work.

Additional Information and Declarations

Competing Interests

Author Contributions

Data Availability

The authors declare there are no competing interests.

Bijendra Khadka and Radhey S. Gupta conceived and designed the experiments, performed the experiments, analyzed the data, prepared figures and/or tables, authored or reviewed drafts of the paper, and approved the final draft.

The following information was supplied regarding data availability:

All of the data are available in the figures and the Figs. S1–S7.

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
