# Peer review of "Conserved molecular signatures in the spike protein provide evidence indicating the origin of SARS-CoV-2 and a Pangolin-CoV (MP789) by recombination(s) between specific lineages of Sarbecoviruses"

_PeerJ, doi:10.7717/peerj.12434_

## Round 0.1 · original submission · Major Revisions

This study is important because it has the potential to add to our understanding of coronavirus recombination in general and the origins of SARS-CoV-2 specifically. However, revision is required in order to present and describe these data clearly. Two of the three reviewers characterized their decisions as "minor revisions required," whereas one reviewer characterized their decision as "major revisions required." After my own review of the manuscript and upon reading their critiques, I would characterize the revisions required as "major revisions" due to the level of rewriting and text reorganization required to address all three reviews combined, as well as Reviewer 2's suggestion to try another [more recent] tool for detection of recombination events.

Two of the three reviewers made suggestions to simplify and decrease text in certain areas of the manuscript. This needs to be done, as well as edits for consistency of labeling virus names and CSI as called out by Reviewer 3 and Reviewer 2, respectively.

Additionally, Reviewer 2 has technical concerns expressed in the Experimental Design section of their critique as well as in the Validity of the Findings section of their critique, that will need to be specifically addressed.

Again, we thank you for your submission and we look forward to reviewing a revised version of this manuscript.

Reviewer 1 ·

Basic reporting

The authors show analysis of conserved inserts and deletions (CSIs) in coronaviruses, in specific, sarbecoviruses. They use this analysis to suggest origins of SARS-CoV-2 and other sarbecoviruses. While investigating the viral protein sequences of the spike and nucleocapsid proteins, they find 17 CSIs and are able to compare these CSIs between lineages of sarbecoviruses and suggest that genetic recombination may have occurred in the evolutionary origin of clusters/clades of sarbecoviruses (ie SARS-CoV-2r).

The paper is clearly written and organized well. The introduction is appropriate for the content of the results and discussion. Figures demonstrate the data well.

Experimental design

Experimental design is well thought out and describes the methodology appropriately. I recommend updating the nomenclature for the variants of concern as the WHO has implemented a new system and the lineage naming for B.1.617 has changed (lines 190-192).

Validity of the findings

Conclusions are well supported and comparisons to other published studies are made where appropriate. Sequences used are properly labeled for reference.

Additional comments

Please correct spelling of Cambodia in line 117. In addition please update citations style of line 105 to match the other citations in the manuscript. Line 394 should state marked '7' and line 399 should state marked '8'. Figures 1 and 9 would benefit from increased resolution. Figure 8 needs to be reformatted for better viewing as the text covers the figure.

Reviewer 2 ·

Basic reporting

In this manuscript, the authors investigate the recombination origins of the coronaviruses and the origins of the SARS-CoV-2 by using phylogenetic analyses and observations of conserved indels. The authors also model indel positions on the structure of spike protein. This work definitely adds to our understanding of coronavirus recombination complexity, however, I have major concerns that would need to be addressed:

Major comment 1: The manuscript is well written but it feels way too long and laborious, with too many figures that show the same concept. There is a great deal of repetition in the written paragraphs as well. The authors intermix background with results quite often. I think that the manuscript can be written in a more concise and simpler way, it is possible to make it half as long as it is now, maybe even shorter. As it is now, it shows the author’s thought process in detail, however, this can all be put in text in a more simple way. I would suggest doing that, as well as moving some of the less important figures into supplemental material.

Minor comment 1: CSI labels in figures and in the text are at times off.
Minor comment 2: Figures 2-6 are unnecessarily large and contain repetitive information, can be made into 1-2 figures with panels instead.
Minor comment 3: I could not properly review figure 8 as the text was scrambled on top of the figure, this figure needs to be made better.
Minor comment 4: I would refrain from numbering paragraphs in Discussion.

Experimental design

Major comment 1: There are plenty of recombination detection strategies out there, whether with or without help of phylogeny. The approach of manually making separate trees for regions of interest in order to detect recombination is quite outdated and may serve as an initial screen. This is also true for pairwise distance comparisons (amino acid or nucleotide, which for obvious reasons would always give very similar results, and did here as well.) I would recommend the authors try to confirm their recombination hypothesis using some of the more recent tools for recombination detection. This may also help to detect where a putative recombination breakpoint would be situated.

Minor comment 1. I would find Figure 1 a lot more informative if all the taxa that was used for inference of the Spike tree were also used in the separate inferences of NTD and RBD.

Validity of the findings

Major comment 1: The authors need to discuss the possibility of indel convergence and reversion, which does happen, and how this would affect their conclusions. The authors highlight that the CSI patterns are explained by recombination since it is the most parsimonious explanation, however, they fail to discuss other possible mechanisms that could give similar results.

Minor comment 1: The statement in line 306 that trees are often not resolved due to the listed variables is not correct. Most often trees are unresolved when there is not enough signal in the data to provide correct relationships between taxa. All the mentioned variables affect the topology of the tree, and less so whether it is resolved or not.
Minor comment 2: Line 273-274 statement that the described clusters and lineages are not supported by good bootstrap is incorrect. As far as I can see from the tree, lineage 3 is supported by 99% bootstrap, Lineage 1 support looks like it is 98% (low image resolution - that should be fixed before publication) , CoVZC is supported by 100% and so is lineage 2. I would not call this low bootstrap support, this is very good support. In addition, I would not consider the branches of the tree short, given the scale of 0.2, from viral evolution perspective.
Minor comment 3: Figure 9 on its own is very unclear, just by looking at it I am having hard time understanding the mechanisms that the authors are trying to show. I do not understand what arrows and x means in the figure. In addition, it seems to me that the authors have placed a putative recombination breakpoint in the figure, right between S1-NTD and S1-CTD, however, there is no support in the article exactly where a breakpoint should be positioned. Also, to me it looks like S1-NTD and S1-CTD lengths differ between panels A and B, although by the number of amino acids they should be of identical length.

·

Basic reporting

The authors did a great job of detailing the complex and rapidly changing landscape of SARS-CoV-2 spike evolution. This is clearly a relevant topic and the authors presented a thorough yet interesting perspective of the ancestry of these conserved indels. I would recommend this paper for publication in PeerJ upon consideration of the following major concerns:

1. I think the authors should reformat and trim down their text in the results section 3.2. While I can’t argue against the importance of any one CSI metric or observation, I think too much text is dedicated to each and in my opinion the reader loses the importance of each in relation to each other. Only at line 576, when figure 9 is referenced, did the author’s interpretation of the CSIs start to come fully into picture for me. I think it would benefit the reader if the 3.2 section were laid out in sub-headers:

1a. S1-NTD, e.g. unless I misinterpret, lines 322-357 and 370-387 are setting the stage for the exact same evolutionary interpretation for CSI 1-6 in S1-NTD.
1b. S1-CTD.
1c. Others of interest – in here, I’d consider stripping the most text in reference to these other CSIs. E.g. when I read lines 358-369 and 411-430 I think juxtaposing them as i.e. “extra CSIs not informing recombination” would help the reader in interpreting the importance of these other CSIs.

I would consider revising the figures to follow this structure as well. E.g. Figure 2a, 2b, and 3b could be combined to highlight the similar interpretation taken from each CSI. Perhaps the actual amino acid sequence could be moved to supplemental and graphical depictions (akin to Figure 9) of the relationships and identified CSIs could be used in place.

Overall, I’d consider a strong editorial eye in terms of excessive text in reference to the CSIs. E.g. Line 579-587 had been stated repeatedly throughout. I’d make every effort to get to the point of line 587 more quickly.

2. The conclusions section is poorly framed. The immediate jump into the importance of this work for diagnostics, therapeutics, and vaccines seems to come from nowhere while the majority of the text has been dedicated to evolutionary dynamics and the impact of recombination. E.g. If the authors intend to emphasize diagnostics, I would introduce this concept more throughout. However, a simple rehash of figure 9 in context with the phylogeny, identity, and protein structure would be sufficient for the conclusion.

Experimental design

No concerns with the experimental design. The authors use the majority of available zoonotic data of related SARS-CoV-2 strains.

Validity of the findings

No comment.

Additional comments

Minor comments:
1. Throughout, the authors use various nomenclature for BatCoV-RaTG13, some with spaces and others with dashes. Please consider standardizing to Line 459 “Bat-CoV-RaTG13”
2. Line 376: I’d recommend the expansion on this point about sequencing error in lieu of more time on diagnostic impacts in the discussion. I’d consider adding a line or two in the discussion to reinforce the potential limitation of this work as it applies to sequencing errors and how bioinformatic pipelines may influence the presence/absence of indels.
3. The authors could bolster their rationale for CSI investigations by referencing the other ways in which to investigate these recombination hypotheses either computationally or through reverse genetic techniques. Presumably the authors could reference the difficulty in wet lab approaches to confirming these recombination hypotheses because of the clear issues regarding gain-of-function experimentation.
4. When the authors reference the CSI they occasionally reference just the number (e.g. Line 655) and other times reference the number and figure (e.g. line 697). I’d recommend following the structure of 697.
5. Figure 8 & 9. Any chance of labeling the CSI number on Figure 9 as well? It would help when reading the text. E.g. place 6, 3, 1, 8, and 9 on the SARS-CoV-2r S1-NTD CSIs. Similarly, there are overlapping formatting issues on my version of Figure 8, but I’d also include the CSI numbers here too.
6. Line 52: I’d describe what RShSTT is in the abstract like you do for CoVZC/PrC31.
7. Line 102-105: difficult sentence structure, consider moving the MERS-CoV fragment to the front of the sentence.
8. Line 112: list the species of RaTG13 isolation.
9. Line 121: “distantly” not “distinctly”.
10. Line 140-142: needs commas.
11. Line 184: change “a” to “the”.
12. Line 189: change “variant of concerns”, to “variants of concern”.
13. Line 328: I think that figure 2 is NTD but authors reference CTD here.
14. Line 409: change to “significance”, not “significances”.
15. Line 462, authors reference C-terminal region of the spike protein (i.e. from aa 320-1265). However, this region is not shown in figure 7.

---

## Round 0.2 · Minor Revisions

Thank you for paying close attention to the three reviewers' comments; they are unanimous in their opinion that the manuscript is improved. There are a few minor details that need addressing; please see Reviewer 1 and Reviewer 3's comments.

Reviewer 1 ·

Basic reporting

The length of the manuscript is greatly improved. The authors have made my recommended changes. Unfortunately, the classification of Delta they made is incorrect. The methods says B.1.617 but is should say B.1.617.2.
B.1.617.1 (Delta) is written in all the figures but the sequence used for analysis ( EPI_ISL_2400521) is B.1.617.2 (Delta). B.1.617.1 is actually the Kappa variant and wasn't used for the analysis. These changes need to be made prior to publication.

Experimental design

The use of RDP5 on default settings seems unwise as it does not account for the S1-NTD sequence region, a region discussed extensively in this manuscript. Perhaps testing other programs would prove more useful for validation of recombination. The resolution of figure S7 is also poor and needs to be improved.

Validity of the findings

No comment.

Additional comments

Minor comment: Please fix the arrow, it is covering the Ssp1 text in Supplemental Figure 5.
Minor comment: Figure legends need to be updated, I noticed a few discrepancies in the main figure legends.
Minor comment: Figure 6 has an incongruent recombination breakpoint label and graphical representation of it. Also the percent identity in A (96.86% as this corresponds to Wuhan-Hu-1 but this an unidentified virus) is a bit misleading, perhaps label it so as to make this clear.

Reviewer 2 ·

Basic reporting

All concerns have been addressed

Experimental design

All concerns have been addressed

Validity of the findings

All concerns have been addressed

Additional comments

The article still feels little too long but I can see that the authors made significant attempts to shorten it and simplify. Current structure is a lot better.

·

Basic reporting

The authors have done a great job polishing this manuscript and for the most part have addressed all of the concerns I had in my first review. I appreciate their effort to synthesize a number of deeply thought out interactions and navigate the reader through the complex scenarios. It is now much more concise, clear, and interpretable. As the authors said, I think their movement of certain figures to the supplement and their modifications to the figures that remain are a big improvement.

Three additional points: 1. I still think there could be some text removed/combined from the discussion where points are reiterated throughout, for example in lines 498-537. 2. I'd consider revising the title more. My opinion would be “Conserved Molecular Signatures in the Spike Protein Provide Evidence Indicating the Origin of SARS-CoV-2 and Pangolin-CoV (MP789) by Recombination(s) between Specific lineages of Sarbecoviruses”. 3. While i recommended that the authors consider discussing other computational techniques to assess recombination I did not necessarily suggest that they ADD analyses to this paper and I'm not immediately sure of the added value in the addition of RDP5 or the sufficient description of that figure. Nevertheless, it does support that recombination occurred and does not take up much text. I'd advise the authors to do one more detailed look at whether they think this analysis is truly needed or whether they just added it to satisfy a reviewer. If the latter then i would remove.

Experimental design

No concerns with the experimental design. The authors use the majority of available zoonotic data of related SARS-CoV-2 strains.

Validity of the findings

No comment.

Additional comments

- In figure 1 and S1, I’d consider adding "RShSTT" into the description on the trees at that clade. It is referred to a number of times in the text and the reader doesn't always know which clade to go back to when it just states "Other Bat-SARS-CoV-2 related CoV strains" on the tree.

- Line 276, spell out your reference to “B-1” when first introducing it here. Presumably the authors are referring to lineage-1?

- 492-496 is a difficult sentence to read. Perhaps break into two sentences?

- Line 550 "...play A critical role...”

---

## Round 0.3 · accepted · Accept

Thank you for addressing the reviewers' comments.